# Unifying Text, Tables, and Images for Multimodal Question Answering

**Haohao Luo[1], Ying Shen[1], Yang Deng[2]\***
[1]School of Intelligent Systems Engineering, Sun Yat-sen University
[2]National University of Singapore
luohh5@mail2.sysu.edu.cn, sheny76@mail.sysu.edu.cn, ydeng@nus.edu.sg

## Abstract

Multimodal question answering (MMQA), which aims to derive the answer from multiple knowledge modalities (*e.g.*, text, tables, and images), has received increasing attention due to its board applications. Current approaches to MMQA often rely on single-modal or bi-modal QA models, which limits their ability to effectively integrate information across all modalities and leverage the power of pre-trained language models. To address these limitations, we propose a novel framework called UniMMQA, which unifies three different input modalities into a text-to-text format by employing position-enhanced table linearization and diversified image captioning techniques. Additionally, we enhance cross-modal reasoning by incorporating a multimodal rationale generator, which produces textual descriptions of cross-modal relations for adaptation into the text-to-text generation process. Experimental results on three MMQA benchmark datasets show the superiority of UniMMQA in both supervised and unsupervised settings.

## 1 Introduction

In typical question answering (QA) systems, answers are typically derived from a single modality, such as an image (Antol et al., 2015), passage (Choi et al., 2018), or table (Oguz et al., 2022), without the need for cross-modality reasoning. However, in real-world scenarios, individuals often rely on multimodal information and reasoning from diverse knowledge sources to arrive at answers. Multimodal question answering (MMQA) (Hannan et al., 2020; Talmor et al., 2021) demands QA systems to perform reasoning across multiple knowledge modalities, including images, structured tables, textual passages, etc.

Preliminary attempts in MMQA (Hannan et al., 2020; Talmor et al., 2021) typically utilize separate QA systems for different knowledge modal-

---

\* Corresponding author.

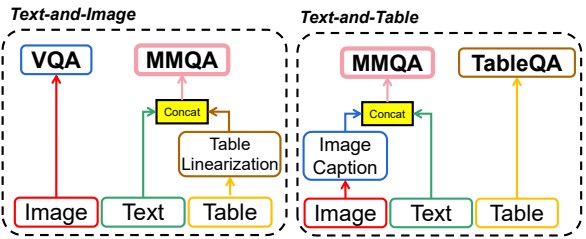

(a) Illustration of existing modality unification methods.

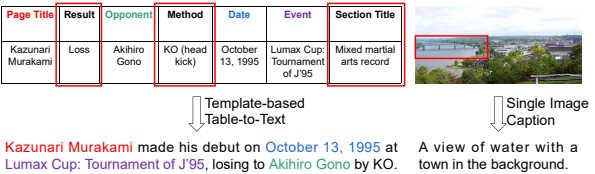

(b) Examples of template-based table-to-text transformation and single image caption.

Figure 1: Illustration of remaining challenges in existing modality unification methods for MMQA.

ities, which failed to bridge the challenging gap between multimodal sources. To integrate information across modalities, various graph structures (Yang et al., 2022a; He and Wang, 2023) are introduced into MMQA systems to build semantic connections between different modalities. However, the graph structure is typically less compatible and practical with language models, and can barely compete with the powerful pre-trained language models (PLMs). To better leverage the knowledge from PLMs, recent studies employ modality unification techniques to transform different modalities into text. For example, tabular data can be either linearized into token sequences (Xie et al., 2022; Liu et al., 2022a) or adapted to pre-defined textual templates (Yoran et al., 2022), while visual data can be summarized into textual descriptions by image caption techniques (Gui et al., 2022; Lin et al., 2022; Yang et al., 2022b; Shao et al., 2023).

Despite the effectiveness of modality unification techniques in handling multimodal inputs, there are several challenges that remain to be tackled

in MMQA: 1) Existing methods only investigate the unification of two modalities, such as text-and-table (Xie et al., 2022; Yoran et al., 2022) or text-and-image (Gui et al., 2022; Lin et al., 2022; Yang et al., 2022b; Shao et al., 2023), while an additional uni-modal QA model is required to handle the rest of questions in MMQA, as shown in Figure 1(a). 2) The process of modality transformation inevitably causes information loss. For example, the position information of each table cell is discarded when adopting template-based table-to-text transformation, as shown in the left example of Figure 1(b). Besides, some important image details will be left out when generating a single image caption, *e.g.*, the bridge in the right example of Figure 1(b). 3) These MMQA approaches using modality unification typically concatenate the unified textual information as the context (Figure 1(a)), while cross-modal reasoning, which attaches great importance in MMQA, is overlooked in these methods.

In the light of these challenges, we propose a novel MMQA framework, namely **UniMMQA**, which unifies three different input modalities into a text-to-text format. In specific, to alleviate the information loss during the modal unification, we first adopt an easy-to-apply decoding strategy to generate informative image captions from the visual data by sampling a diverse set of image descriptions instead of only taking the greedy one. Meanwhile, the tabular data is linearized into textual sequence with position tokens. Then a cross-modal rationale generator is adapted to produce explicit textual reasoning descriptions about the interrelations across different modalities. All the multimodal knowledge and the cross-modal rationale are represented in the text form, which are ultimately applied to a text-to-text generation with PLMs. The main contributions of this work can be summarized as follows:

- We propose a novel framework, namely **UniMMQA**, to unify text, tables and images in MMQA into a text-to-text format for PLMs, via position-enhanced table linearization and diversified image captioning.

- To enhance the cross-modal reasoning over texts, we prompt a multimodal rationale generator to produce textual cross-modal relation descriptions for adapting into the text-to-text generation.

- Experimental results on three MMQA benchmark datasets show that UniMMQA outperforms existing MMQA methods and effectively takes advantage of PLMs. Our code will be released via `https://github.com/Luohh5/UniMMQA`.

## 2 Related Works

**Multimodal Question Answering** Evolving from visual question answering (VQA) (Antol et al., 2015) that aims to answer questions from image-only inputs, knowledge-based VQA studies (Marino et al., 2019; Shah et al., 2019) expand the scope to involve both textual and visual knowledge. Another line of studies (Wang et al., 2022) focuses on QA over a hybrid context of tabular and textual data (Chen et al., 2020b, 2021). Due to the multimodal nature of information flow in real-world applications, researchers (Hannan et al., 2020; Talmor et al., 2021; Chang et al., 2022; Li et al., 2022b) emphasize the importance of answering questions that require information across multiple modalities, including text, tables, and images, which is typically referred as multimodal question answering (MMQA). MMQA has been widely used in various real-world applications, such as finance (Zhu et al., 2021; Deng et al., 2022a), e-commerce (Deng et al., 2023, 2022b, 2020), science (Lu et al., 2022; Xu et al., 2021b), and more. Early studies (Hannan et al., 2020; Talmor et al., 2021) decompose MMQA into three single-modal QA models. To align different modalities, some latest studies employ graph structures (Yang et al., 2022a; He and Wang, 2023) to enhance the cross-modal interaction.

**Modality Unification** Pre-trained language models (PLMs) show exceptional proficiency in handling text-to-text problems (Raffel et al., 2020), which can be further adapted into large language models (Zhao et al., 2023). Therefore, a mainstream approach for handling multimodal tasks is to unify different modalities into text. As for knowledge-based VQA, a recent trend is to leverage image caption techniques for unifying the information across textual and visual modalities, which either generates captions from the whole image (Gui et al., 2022; Lin et al., 2022) or utilizes the information of object regions for image captioning (Yang et al., 2022b; Shao et al., 2023). As for table-and-text hybrid QA, some latest studies leverage either table linearization techniques (Xie et al., 2022; Liu et al., 2022a) or template-based table-to-text approaches (Yoran et al., 2022) to combine tabular and textual data. However, there has been

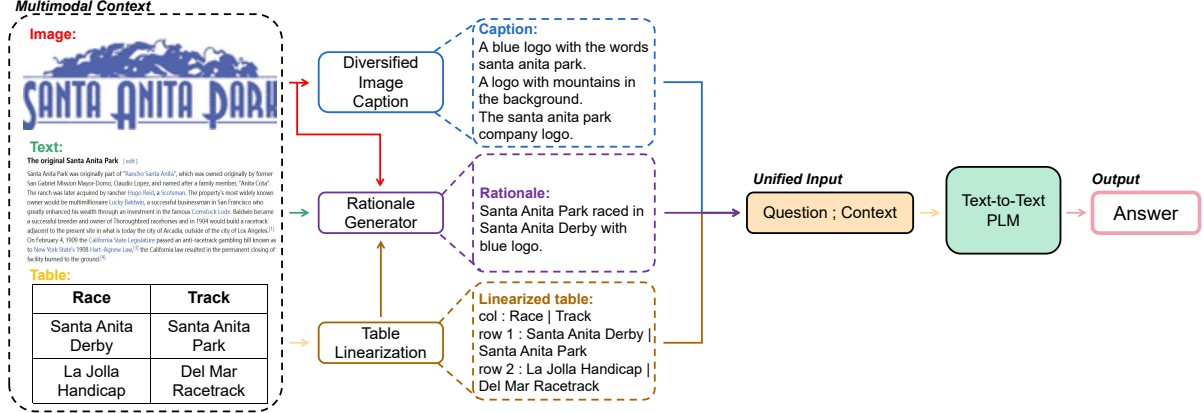

Figure 2: The overall framework of UniMMQA.

relatively little work on unifying text, tables and images for multimodal question answering. A contemporary work (Yu et al., 2023) also proposes to transform the images and tables into unified language representations for solving MMQA as a textual QA problem. Nevertheless, we further mitigate the information loss issues during the modality unification in MMQA.

**Image Caption** Image caption (Stefanini et al., 2023) aims to generate a natural language caption to summarize the objects and their attributes information in a given image. In recent years, several studies utilize powerful visual-language pre-training (VLP) models to realize the captioning process (Radford et al., 2021; Li et al., 2022a; Yuan et al., 2023). Moreover, some researchers design advanced methods for generating more diverse and informative image captions (Chen et al., 2019; Zhao et al., 2019; Mahajan and Roth, 2020; Xu et al., 2021a). In this work, we investigate diversified image caption techniques to alleviate the information loss during modality unification.

## 3 UniMMQA Framework

Given a question $Q$ and corresponding knowledge context $C = \{I, T, P\}$, where $I$ represents the image, $T$ represents the table and $P$ represents textual passage, UniMMQA aims to generate an appropriate answer $A$. The overview of UniMMQA is illustrated in Figure 2.

### 3.1 Diversified Image Caption

In order to perform the downstream task in text space, we convert the image knowledge into text. Specifically, we employ two conversion strategies: optical character recognition (OCR) and image caption. Firstly, according to Jain et al. (2021), we utilize an off-the-shell OCR model[1] to extract all explicit text in the images. Secondly, following BLIP (Li et al., 2022a), we apply a state-of-the-art image captioning model with some modifications to transform the image into caption text from noisy web data. In order to diversify the caption generation, we adopt an easy-to-apply decoding strategy by sampling a diverse set of image descriptions instead of taking the greedy one (Vijayakumar et al., 2016). The sampling strategy picks the word according to conditional probability distribution:

$$I_t^{text} \sim P(I^{text}|I_{1:t-1}^{text}), \quad (1)$$

where $I_t^{text}$ denotes the word in the sampling pool and $t$ denotes the next picking. In specific, we jointly employ Top-K (Fan et al., 2018) and Top-p (Holtzman et al., 2020) sampling. In Top-K sampling, the $K$ most likely next words are filtered and the probability mass is redistributed for picking the next word according to their cumulative probability:

$$\sum\nolimits_{I_t^{text} \in V_K} P(I_t^{text}|I_{1:t-1}^{text}), \quad (2)$$

where $V$ denotes the sampling pool. In Top-p sampling, the smallest possible set of words whose cumulative probability exceeds the probability $p$ are filtered for the next word picking, which can be described as:

$$\sum\nolimits_{I_t^{text} \in V_x} P(I_t^{text}|I_{1:t-1}^{text}) \geq p, \quad (3)$$

However, Top-p sampling only filters a few words in the sampling pool when the next word seems more predictable, which means it's more elegant than Top-K sampling. To handle this, we utilize

---

[1] https://github.com/JaidedAI/EasyOCR

Top-p in combination with Top-K as our decoding strategy to avoid very low-ranked words. Our strategy is then the following:

$$\sum\nolimits_{I_t^{text} \in V_x} P(I_t^{text}|I_{1:t-1}^{text}) \geq p, x \geq K, \quad (4)$$

where $x$ is set to $x \geq K$ to prevent filtering too few words in the sampling pool. To sum up, we feed the image into a vision transformer (Dosovitskiy et al., 2021) to obtain patch-level features, which are then fed into a decoder to generate the image caption based on Top-K and Top-p sampling strategy.

$$I^{text} = F_{caption}(I, K, p), \quad (5)$$

where $F_{caption}$ is a vision-language pre-trained model consists of encoder and decoder.

## 3.2 Table-to-Text Transformation

Besides to image, structured table also need to converted into text. A commonly used conversion method is linearization. In this work, we employ position-enhanced encoding linearization (Liu et al., 2022b). Specifically, we concatenate all elements on the same row and different elements are separated by " | ". All the rows including the header are then concatenated into a long passage delimited by predefined separator "header :" or "row : $x$" where $x$ denotes the row id. Our linearized table can be represented as:

$$T^{text} = \text{“ header} : h_1 \mid ... \mid h_N$$
$$\text{row } 1 : r_{11} \mid ... \mid r_{1N}$$
$$...$$
$$\text{row M} : r_{M1} \mid ... \mid r_{MN}\text{”}$$

where $M$ and $N$ denote the number of rows and columns respectively. Therefore, we unify different modalities of the context $C$ into textual sequence $C^{text} = \{I^{text}, T^{text}, P\}$.

## 3.3 Multimodal Rationale Generation

In addition to unifying input modalities for text-to-text format, we generate intermediate reasoning descriptions as the rationale to seek the connection and relationship between different modalities. Following Zhang et al. (2023), we fine-tune a generative PLM (*e.g.*, T5) on ScienceQA benchmark as a rationale generator, which generates the rationale based on visual features and textual inputs. Specifically, we feed the image to CLIP (Radford et al., 2021) to extract vision features and we feed

text and linearized table to the language encoder to extract language representations by the following functions:

$$I_{feature} = F_v(I), \quad (6)$$

$$P_{feature} = F_l(T^{text}, P), \quad (7)$$

where $F_v$ denotes vision extraction model CLIP and $F_l$ denotes the language encoder implemented as a Transformer model. Then the vision features and language representations are fused to encode their joint features. The joint features are subsequently fed into decoder to generate rationale, which describes the interrelations across different modalities. The rationale is obtained by:

$$R = F_r(I_{feature}, P_{feature}), \quad (8)$$

where $R$ denotes the rationale and $F_r$ denotes the rationale generation model. After incorporating the natural language rationale into the textual input to enhance the cross-modal interaction, *i.e.*, $C^{text} = \{I^{text}, T^{text}, P, R\}$, we concatenate it with $Q$ as an unified input sequence $X = \{Q_1, Q_2, ..., Q_n, C_1^{text}, C_2^{text}, ..., C_m^{text}\}$, where $n$ and $m$ represent the max length of question sequence and context sequence respectively.

## 3.4 Sequence-to-Sequence Training Procedure

After unifying all modalities to a textual input sequence $X$, we fine-tune a generative PLM (*e.g.*, T5) as our QA model which is defined as:

$$Y = F_{qa}(X), \quad (9)$$

where $F_{qa}$ is the QA model and $Y$ denotes the prediction sequence. When fine-tuning on MMQA task for the answer generation stage, we minimize the negative log-likelihood loss $\mathcal{L}_{NLL}$ averaged over tokens in each batch as our training objective:

$$\mathcal{L}_{NLL} = -\frac{1}{L}\sum_{l=1}^{L} \overline{y_l} \log\left(\frac{\exp(y_l)}{\sum_i^l \exp(y_i)}\right), \quad (10)$$

where $L$ is the max length of output sequence, $\overline{y_l}$ and $y_l$ denote the $l$-th token in gold answer and prediction sequence respectively. Furthermore, we employ prefix-tuning as our fine-tuning strategy, which adds a task-specific prefix (*e.g.*, MMQA) to the input sequence as a prompt.

| Datasets | Split | | | Modality | | |
|---|---|---|---|---|---|---|
| | Train | Dev | Test | Image | Table | Text |
| ManymodelQA | 2,036 | 3,055 | - | 2,873 | 3,528 | 3,789 |
| MultimodalQA | 23,817 | 2,442 | 3,660 | 57,058 | 10,042 | 218,285 |
| MMConvQA | 4,582 | 581 | 590 | 57,058 | 10,042 | 218,285 |

Table 1: Dataset statistics of MultimodalQA, ManymodalQA, and MMConvQA.

# 4 Experiments

## 4.1 Experimental Setups

**Datasets & Evaluation Metrics** We conduct the experiments on three public benchmark datasets, including ManymodalQA (Hannan et al., 2020), MultimodalQA (Talmor et al., 2021), and MMConvQA (Li et al., 2022b). ManymodalQA contains 10,190 questions: 2,873 images, 3,789 text and 3,528 tables, which have been split into train/dev sets. MultimodalQA consists of 29,918 question-answer pairs cross image, table and text modalities, 35.7% of which require cross-modal reasoning. Derived from MultimodalQA, MMConvQA is a multimodal conversational question answering dataset with 1,179 conversations and 5,753 question-answering pairs. There are 218,285 passages, 10,042 tables and 57,058 images in MMConvQA, and about 24.4% of conversations require reasoning cross three modalities. The statistics of these datasets are presented in Table 1. Following previous studies, we adopt Exact Match (EM) and F1 as the evaluation metrics for all datasets.

**Baselines** We compare UniMMQA with the state-of-the-art methods in each benchmark, including Implicit-Decomp (Talmor et al., 2021), MMQA-T5 (Yoran et al., 2022), PReasM (Yoran et al., 2022), SKURG (Yang et al., 2022a), MGT (He and Wang, 2023), ManymodalQA (Hannan et al., 2020), ORConvQA (Qu et al., 2020), MAE (Li et al., 2022b), Solar (Yu et al., 2023). Detailed descriptions of each baseline are provided in Appendix A. Our default configuration utilizes T5 model as backbone. Specifically, we employed different sizes of T5 model to evaluate our performance, denoted as **UniMMQA (T5-Base)**, **UniMMQA (T5-Large)** and **UniMMQA (T5-3B)**.

**Implementation Details** We use three sizes of T5 as our backbone language models, including T5-Base, T5-Large and T5-3B. Besides, we set the max length of unified input sequence and output sequence to 1024 and 128 respectively. The beam

| Method | Dev | | Test | |
|---|---|---|---|---|
| | EM | F1 | EM | F1 |
| Implicit-Decomp | 48.8 | 55.5 | 49.3 | 55.9 |
| MMQA-T5-Large | 57.9 | 64.3 | 57.0 | 63.4 |
| PReasM-Large | 59.0 | 65.5 | 58.3 | 64.6 |
| SKURG | 59.4 | 63.8 | - | - |
| MGT | 52.1 | 57.7 | - | - |
| Solar | 59.8 | 66.1 | - | - |
| UniMMQA (T5-Base) | 67.9 | 74.0 | 67.0 | 72.9 |
| UniMMQA (T5-Large) | 71.3 | 77.1 | 70.1 | 75.8 |
| UniMMQA (T5-3B) | 75.5 | 81.7 | 73.7 | 80.1 |

Table 2: Experimental results on MultimodalQA.

size of the answer generating process is set to 4. For training, we use batch size 4 for T5-Base and use batch size 2 for T5-Large and T5-3B due to CUDA memory. We employ the Adafactor optimizer for T5-Base and T5-Large, and AdamW for T5-3B with the 5e-5 initial learning rate and linear decay of the learning rate. The max training epoch is set to 400 for all datasets. We evaluate on the development set for every 500 steps and use the average development set metric for the best checkpoint selection. As for the image caption module, we jointly adopt Top-K and Top-p sampling as decoding strategy in caption generating. The sampling pool size $K$ and the probability threshold $p$ are set to $K = 50$ and $p = 0.9$ respectively. The number of generated sequences per image is set to $N = 3$ for more diversified image captions.

## 4.2 Overall Performance

**MultimodalQA** Table 2 presents a comparison of the performance of UniMMQA on the MultimodalQA dataset with previous state-of-the-art models. Among the baselines, MMQA-T5 and PReasM, which unify the textual and tabular modalities, largely outperform Implicit-Decomp which adopts separated uni-modal and bi-modal QA models. Despite the improvement over Implicit-Decomp, SKURG and MGT, which employ graph structures to enhance the interactions between different modalities, fail to compete with MMQA-T5 and PReasM. These results indicate the effectiveness of modality unification on the MMQA task. Furthermore, the proposed UniMMQA method surpasses all the strong baselines on both EM and F1 by unifying all three modalities of MMQA. In addition, the performance of UniMMQA also grows with the size of the backbone PLM, which shows the superiority of the proposed method to be adap-

| Method | EM | F1 |
|---|---|---|
| Most Common | 2.4 | - |
| USE | 5.6 | - |
| Voting | 21.1 | - |
| ManymodalQA | 39.7 | - |
| ManymodalQA (w/ Oracle) | 46.3 | - |
| UniMMQA (T5-Base) | 45.4 | 45.7 |
| UniMMQA (T5-Large) | 50.0 | 50.4 |
| UniMMQA (T5-3B) | 52.1 | 52.4 |

Table 3: Experimental results on ManymodalQA.

| Method | Dev | | Test | |
|---|---|---|---|---|
| | EM | F1 | EM | F1 |
| ORConvQA | 1.2 | 3.0 | 1.1 | 1.9 |
| ManymodelQA | 2.3 | 0.7 | 1.8 | 1.0 |
| MAE | 19.8 | 26.8 | 22.0 | 28.3 |
| Solar | 56.8 | 62.5 | 57.3 | 64.6 |
| UniMMQA (T5-Base) | 57.8 | 64.7 | 59.2 | 64.9 |
| UniMMQA (T5-Large) | 62.3 | 69.0 | 63.6 | 70.0 |
| UniMMQA (T5-3B) | 65.8 | 71.4 | 66.7 | 72.6 |

Table 4: Experimental results on MMConvQA.

tive to a variety of powerful PLMs.

**ManymodalQA**    We also conducted experiments on ManymodalQA dataset and the results are presented in Table 3. Compared with heuristic methods, ManymodalQA achieves much better performance by classifying the question types for different uni-modal QA models. With oracle question type labels (w/ Oracle), ManymodalQA further improves the performance, indicating the importance of identifying the modalities of the required information in MMQA. The results demonstrate that UniMMQA outperforms almost all baseline models on the ManymodalQA dataset, but the score of UniMMQA (T5-Base) is lower than ManymodalQA (w/ Oracle).

**MMConvQA**    Table 4 summarizes the results for MMConvQA dataset, in comparison with previous SOTA. We can see that UniMMQA achieves huge improvements and outperforms the best baseline MAE by at least 39.2 EM scores and 37.0 F1 scores. This also indicates that UniMMQA can be flexibly and effectively applied to different types of MMQA problems, such as in the form of conversations.

**Overall**    UniMMQA achieves new state-of-the-art results on all MultimodalQA, ManymodalQA and MMConvQA datasets. Compared with existing methods that decompose MMQA into three single-modal models, *e.g.*, Implicit-Decomp and ManymodalQA, the results show that unifying input modalities performs better. Besides, compare with SKURG and MGT that utilize graph structures to enhance the cross-modal interaction, our multimodal rationale generator is more effective and contributes to the superior results.

### 4.3 Ablation Study

We perform ablation studies to investigate the effects of the proposed approaches in terms of model

| Method | T5-Base | | T5-Large | | T5-3B | |
|---|---|---|---|---|---|---|
| | EM | F1 | EM | F1 | EM | F1 |
| UniMMQA | **67.9** | **74.0** | **71.3** | **77.1** | **75.5** | **81.7** |
| w/o Prefix-tuning | 67.6 | 73.7 | 71.0 | 76.8 | 75.3 | 81.4 |
| w/o Rationale | 67.9 | 73.9 | 71.3 | 77.0 | 75.4 | 81.6 |
| w/o Top-k sampling | 67.7 | 72.9 | 71.1 | 76.1 | 75.2 | 80.7 |
| w/o Top-p sampling | 67.2 | 73.1 | 70.9 | 76.4 | 75.3 | 81.2 |
| w/ Beam search | 66.1 | 72.6 | 69.8 | 75.6 | 74.5 | 80.6 |
| w/ Greedy decode | 66.8 | 72.1 | 71.1 | 75.4 | 75.1 | 80.5 |
| w/ Template-based Table | 41.6 | 50.9 | 47.7 | 56.5 | 52.1 | 60.0 |

Table 5: Ablation study on MultimodalQA (Dev set).

fine-tuning, rationale generation, image captioning, and table linearization, as presented in Table 5. There are several notable observations as follows:

- When dropping the prefix-tuning strategy, UniMMQA uses a common fine-tuning strategy on down-stream tasks without adding a task-specific prefix into the input sequence, whose performance drops by 0.2-0.3 EM and F1 points.

- When we drop the rationale in input sequence, the overall performance declines a little, which demonstrates that adding the interrelations across input modalities to input could indeed benefit enhancing cross-modal reasoning. Although there is a weak correlation between most images and text in MultimodalQA, our rationale generator performs well in generating relation descriptions between other highly correlated images and text. We show a corresponding example in Figure 6.

- As for image captioning, when we drop Top-K or Top-p sampling, we can see declines in both situations. When replacing with beam search(Shao et al., 2017) or greedy decode(Vijayakumar et al., 2016), the performance drops about 2%, which further verifies the effectiveness of jointly utilizing Top-K and Top-p sampling to diversify the generated image captions.

| Prompt | LLaMA-7B | | Vicuna-7B | | LLaMA-13B | | Vicuna-13B | |
|---|---|---|---|---|---|---|---|---|
| | EM | F1 | EM | F1 | EM | F1 | EM | F1 |
| Text + Linear. Table + Image (MiniGPT-4) | - | - | 2.3 | 13.3 | - | - | 1.2 | 14.1 |
| Text + Template. Table + Image Caption | 0.0 | 11.3 | 6.8 | 16.8 | 1.4 | 12.1 | 1.8 | 16.7 |
| Text + Linear. Table + Image Caption | 0.0 | 14.9 | 7.4 | 19.3 | 1.6 | 17.5 | 2.2 | 23.3 |
| Text + Linear. Table + Diversified Image Caption | 0.0 | 15.0 | 7.6 | 19.3 | 2.4 | 19.0 | 2.4 | 23.7 |
| Text + Linear. Table + Diversified Image Caption + Rationale | 0.0 | **15.3** | 7.8 | **19.4** | **2.4** | **19.2** | 2.6 | **24.1** |

Table 6: Zero-shot performance with LLMs on MultimodalQA.

| Dataset | Model | Image | | Table | | Text | | Overall | |
|---|---|---|---|---|---|---|---|---|---|
| | | EM | F1 | EM | F1 | EM | F1 | EM | F1 |
| MultimodalQA | UniMMQA (T5-Base) | 66.7 | 69.2 | 65.5 | 71.4 | 68.3 | 76.4 | 67.9 | 74.0 |
| | UniMMQA (T5-Large) | 69.8 | 72.2 | 69.9 | 75.6 | 71.9 | 80.0 | 71.3 | 77.1 |
| | UniMMQA (T5-3B) | 73.8 | 76.7 | 73.4 | 79.7 | 76.2 | 84.1 | 75.5 | 81.7 |
| ManymodalQA | UniMMQA (T5-Base) | 46.6 | 46.9 | 60.7 | 61.1 | 30.2 | 30.4 | 45.4 | 45.7 |
| | UniMMQA (T5-Large) | 48.5 | 48.6 | 67.5 | 68.2 | 34.9 | 35.1 | 50.0 | 50.4 |
| | UniMMQA (T5-3B) | 49.8 | 50.2 | 58.0 | 58.3 | 40.9 | 41.3 | 52.1 | 52.4 |
| MMConvQA | UniMMQA (T5-Base) | 73.2 | 75.5 | 33.5 | 40.5 | 66.8 | 76.3 | 57.8 | 64.7 |
| | UniMMQA (T5-Large) | 73.2 | 75.4 | 38.9 | 46.8 | 73.3 | 81.7 | 62.3 | 69.0 |
| | UniMMQA (T5-3B) | 73.3 | 75.5 | 41.9 | 47.9 | 77.8 | 85.1 | 65.8 | 71.4 |

Table 7: Detailed performance in terms of different modalities.

• For table linearization, when we use template-based encoding (Chen et al., 2020a; Su et al., 2021) in place of position-enhanced encoding, we convert structured table to natural language. The results show that there is a sharp decline for all model sizes, demonstrating that the template-based conversion to natural language casts a negative impact on the MMQA task due to the diversity and differences of table formats and content.

## 4.4 Zero-shot Setting with LLMs

In order to testify the effectiveness of the proposed framework with the use of LLMs, we evaluate the zero-shot performance of three different open-source LLMs, including LLaMA (Touvron et al., 2023), Vicuna (Chiang et al., 2023), and MiniGPT-4 (Zhu et al., 2023). Due to the length limits in these LLMs (*e.g.*, 2048 tokens for Vicuna), the few-shot in-context learning is impractical where the context in MMQA is much longer than other tasks. Without few-shot samples, the format of the generated answers is typically a complete sentence, instead of a short phrase or a text span as the reference answer. Therefore, we observe that the exact match (EM) score is unreliable, and here we mainly discuss the results in terms of the F1 score.

Table 6 summarizes the zero-shot performance with LLMs on the MultimodalQA dataset. There

are several notable observations: 1) Multimodal LLMs (*e.g.*, MiniGPT-4) for encoding the original image, which demonstrates superior performance on the VQA tasks, fall short of handling the MMQA task, compared with their base LLMs with image captions. 2) Due to the instruction tuning, Vicuna performs much better than LLaMA. 3) The position-enhanced table linearization outperforms the template-based one to a great extent. 4) Both the proposed diversified image caption technique and the generated rationale further improve the performance of UniMMQA. Overall, these results validate the applicability of the proposed UniMMQA framework in the era of LLMs.

## 4.5 Further Analysis

**Analysis of Different Modalities** We analyze the performance in terms of questions requiring the information from different modalities, and the results are summarized in Table 7. As shown in the results on MultimodalQA datasets, the performance of UniMMQA on image-based and table-based questions is observed to be lower compared to the overall performance, whereas it performs better on text-based questions. This disparity can be attributed to the information loss that may occur during the conversion of images and tables into text, as well as the inherent challenges in effectively incorporating

visual and tabular data into the model's reasoning process. Besides, the EM and F1 on text-based and image-based questions are lower than overall performance but those on table-based questions are higher on ManymodalQA dataset. The reason is that the tables provided in ManymodalQA only contain a few rows and its passages are too lengthy and complex. Therefore, UniMMQA is hard to make well reason from passage context but does well in table context on ManymodalQA dataset. Conversely, the performance on table-based questions is much lower than the overall performance in MM-ConvQA, while it performs better on image-based and text-based questions. This indicates that it is challenging to handle conversational question answering over tables. Overall, we conclude that the performance on the MMQA task is largely affected by the complexity of the context from each modality. Therefore, it is worth studying approaches to effectively unifying different modalities.

**Analysis of Number of Sampled Image Captions**
In order to generate more diversified image caption, we set different numbers of sampled image caption $N$ and pick the top $N$ sequences with the highest cumulative probability. Figure 3 shows how the number of sampled image caption affects the overall performance. We apply our best model trained on MultimodalQA and test it with varying $N$. As shown, the EM and F1 both increase along with the number of sampled image caption $N$ from 65.8 to 67.9 and 72.1 to 73.9 as $N$ increases from 1 to 3. A larger $N$ increases the caption diversity that each image generates more than one corresponding caption. However, after $N$=3, the testing performance decreases along with the growing number of $N$. This is because setting too large $N$ resulting in generating lengthy caption, which is hard for QA model making well reasoning.

## 4.6 Case Study

To evaluate our modality unification module, we present generated examples from MultimodalQA dataset in Figure 4 to Figure 6. Figure 4 shows that when we drop out our diversified image captioning strategy, the caption model fails to find the *bridge* in the given image and thus generates the wrong answer. When adding the strategy, the three generated captions contain more detailed information in the image and thus answer correctly. Figure 5 shows how different table linearization strategy affects the answer generating. When we

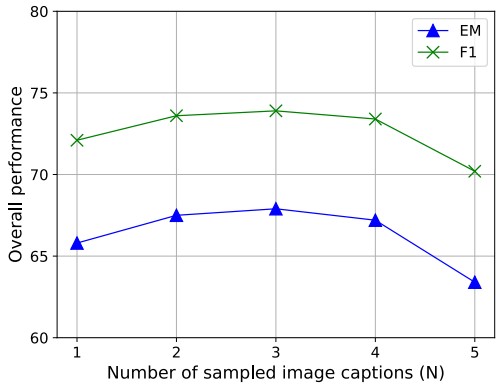

Figure 3: The overall performance with varying number of sampled image caption.

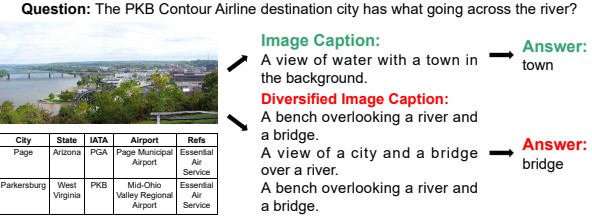

Figure 4: Case study in terms of different image caption methods.

use template-based encoding, the predefined template only selects columns *Position*, *Country* and *Athlete* to generate natural language and discards column *Height* which is mentioned in question, thus resulting in wrong answer generation. And when using position-enhanced encoding, all the cells are converted into text without any information loss, thus the model could find the correct cell and select the right answer. In the example from Figure 6, when we add the multimodal rationale generator, the model could connect *New York Yankees* passage with the logo to produce their relation description and generate the right answer. When dropping the rationale, there is no connection between the passage and the image, which make QA model hard to make good reasoning. Overall, we conclude that adding our modality unification module indeed effectively unifies different modalities and yields decent performance.

## 5 Conclusion

In this paper, we present a novel MMQA framework called UniMMQA, which unifies text, tables and images into a text-to-text format for PLMs. Specifically, we first utilize an effective decoding strategy to generate diversified and informative image captions from the visual data. And we linearize the tabular data into textual sequence with

**Question:** Which blonde haired athlete competed in the IAAF Golden League of 2007 in the Women's High Jump with a height of 1.91?

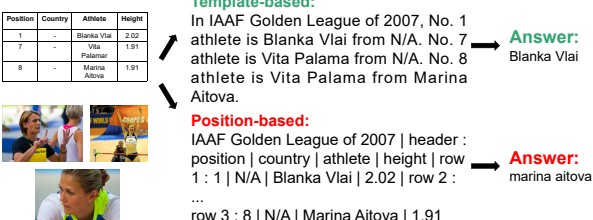

**Template-based:**
In IAAF Golden League of 2007, No. 1 athlete is Blanka Vlai from N/A. No. 7 athlete is Vita Palama from N/A. No. 8 athlete is Vita Palama from Marina Aitova. → **Answer:** Blanka Vlai

**Position-based:**
IAAF Golden League of 2007 | header : position | country | athlete | height | row 1 : 1 | N/A | Blanka Vlai | 2.02 | row 2 : ...
row 3 : 8 | N/A | Marina Aitova | 1.91 → **Answer:** marina aitova

Figure 5: Case study in terms of different table linearization methods.

**Question:**
Among MLB teams that are currently affiliated with the International League, when did the team with a baseball bat on their logo last win a World Series tournament?

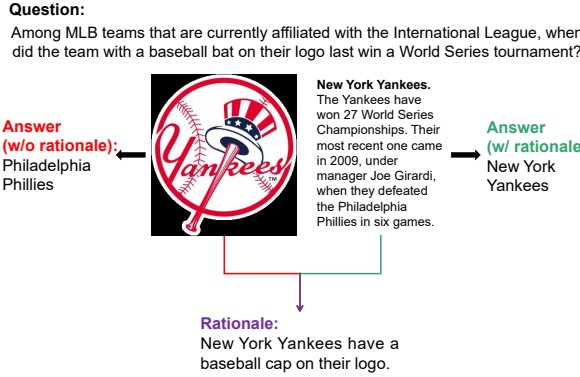

**New York Yankees.**
The Yankees have won 27 World Series Championships. Their most recent one came in 2009, under manager Joe Girardi, when they defeated the Philadelphia Phillies in six games.

**Answer (w/o rationale):**
Philadelphia Phillies

**Answer (w/ rationale):**
New York Yankees

**Rationale:**
New York Yankees have a baseball cap on their logo.

Figure 6: Case study in terms of with and without rationale generation.

position tokens. Meanwhile, a cross-modal rationale generator is adapted to produce explicit textual descriptions about the interrelations across different modalities. Our experiments on three MMQA benchmark datasets show that UniMMQA outperforms many MMQA methods and effectively take advantage of PLMs.

## Acknowledgements

This work was supported in part by the National Natural Science Foundation of China under Grant 61602013.

## Limitations

**Modality Unification** Although using our image captioning techniques could generate informative and diversified image captions, information gap still exists between image and caption. Specifically, our image model lacks the ability to extract very detailed information in image (*e.g.* the number of objects), thus resulting in wrong answering when facing some corresponding questions. Meanwhile, although using position-enhanced table linearization could convert all content into text without any information loss, it also increases the complexity of input sequence and largely affects the overall performance. Therefore, it is worth studying approaches to balance the informativeness and complexity during the modalities conversion in our future work.

**Hallucination Issues** Another limitation of UniMMQA is that our image caption and rationale generation module may suffer from the typical flaw of hallucination issues, *i.e.*, generating fabricated descriptions that is irrelevant to image or text. Fabricated caption and rationale would mislead the reasoning and result in incorrect answering. Our current approach has not taken this issue into consideration, leaving room for future research to explore and address this aspect in greater detail.

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

## A  Detailed Descriptions of Baselines

We detailedly introduce the compared baseline methods as follows:

- Implicit-Decomp (Talmor et al., 2021), which consists of several single-modality and multi-modality QA models, predicts a program that specifies the required reasoning steps over different modalities, and executes the program with different QA models.

- MMQA-T5 (Yoran et al., 2022) first fine-tunes T5 (Raffel et al., 2020) on the MMQA task, with the linearized tables and text as the input. To handle image-based questions, Yoran et al. (2022) route those questions to Implicit-Decomp and the others to the fine-tuned T5.

- PReasM (Yoran et al., 2022) further fine-tunes T5 with different reasoning skills with synthetic tabular data, and then adopts the same question routing approach for handling image-based questions as MMQA-T5.

- SKURG (Yang et al., 2022a) utilizes the knowledge graph to integrate the multimodal inputs for modeling interdependent reasoning steps.

- MGT (He and Wang, 2023) also employs graph structure to model interactions between different modalities. MGT utilizes Transformer as the backbone and combines multimodal graph learning from unstructured data with Transformers.

- ManymodalQA (Hannan et al., 2020) utilizes a question-type classifier to determine the modality which the question belongs and directed the question and the context into the corresponding uni-modal QA models.

- ORConvQA (Qu et al., 2020) is an open-retrieval conversational QA model, which consists of a learnable retriever, a reranker and a reader all based on Transformers.

- MAE (Li et al., 2022b) is a multimodal conversational QA model (MMCoQA), which divides the MMCoQA task into three steps: conversational question understanding, multimodal evidence retrieval, and adaptive answer extraction.

- Solar (Yu et al., 2023) consists of two parts: a unified language representation and a unified QA model. The former is responsible for transforming images, tables and texts in different modalities into language representation. The latter generates an answer through three steps of retrieval, ranking and decoding in the language space.