# OpenReview forum: "Unifying Text, Tables, and Images for Multimodal Question Answering"
_EMNLP/2023/Conference — EMNLP 2023 Findings_

### Official Review · Reviewer_smpP · 2023-08-04

**Typos Grammar Style And Presentation Improvements:** N/A
**Soundness:** 2

**Excitement:**

2: Mediocre: This paper makes marginal contributions (vs non-contemporaneous work), so I would rather not see it in the conference.

**Missing References:**

N/A

**Paper Topic And Main Contributions:**

This paper studies multi-modal QA, that given input modalities (text, images, tables) the model is asked to answer questions. Their main contribution is to unify three different input modalities into text, by employing position-enhanced table linearization and diversified image captioning techniques.

**Questions For The Authors:**

1. As mentioned in the limitation section, "our image model lacks the ability to extract very detailed information in image", using image captioning model instead of raw visual features will inherently lose fine-grained information. As your main method are fine-tuning T-5 models, have you compared with plugging-in a visual encoder (or a frozen visual encoder) and fine-tuning it?
2. Is figure-2 a good example of rationales? "Look at the picture of the Santa Anita Park logo." I don't see why this rationale will help downstream QA.

**Reasons To Accept:**

1. Paper is structured.
2. They conducted comprehensive experiments and ablation studies.

**Reasons To Reject:**

1. The technical novelty and contribution are very limited. They convert images into text with multiple captions, and tables into text by rows and row id. Neither provide new insights. In the literature of vision-language and table-to-text NLP, there are more advanced methods to use to convert image features and tables.
2. They have another module which generates the "rationale" based on visual features and textual inputs. by fine-tuning a generative PLM on ScienceQA benchmark as a rationale generator. But again this is following previous work (Zhang et al 2023). I am also not sure to what degree one can claim this as cross-modal reasoning.

**Reproducibility:**

3: Could reproduce the results with some difficulty. The settings of parameters are underspecified or subjectively determined; the training/evaluation data are not widely available.

**Reviewer Confidence:**

4: Quite sure. I tried to check the important points carefully. It's unlikely, though conceivable, that I missed something that should affect my ratings.

---

> ### Author Rebuttal · Authors · 2023-08-28
>
> I would like to express my gratitude for your time and effort in reviewing our paper. We appreciate your valuable feedback and suggestions, which have prompted us to revisit our work and address the concerns raised in your evaluation. In this rebuttal, we aim to clarify certain points and provide further justification for the significance and contributions of our research.
>
> 1. Regarding the novelty of the proposed method, we would like to clarify as follows.
>
> As for the image-to-text convertion, we don’t just generate multiple captions but also improve the decoding strategy of the existing advanced caption model to diversify the caption generation. Specifically, we utilize Top-p in combination with Top-K as our decoding strategy and set K = 50 and p = 0.9 to reduce the multiple generated captions similarity and promote caption diversification. This strategy effectively addresses the concern of generating repetitive or similar captions.
>
> As for the table-to-text convertion, we explored the use of template-based decoding as an alternative to our position-enhanced decoding approach. However, our experimental results, as shown in Table 5 of our paper, demonstrate that template-based decoding yielded inferior performance. In contrast, our position-enhanced decoding method not only achieves better results but also exhibits high efficiency, making it a practical and reproducible solution for table-to-text conversion. Given these compelling outcomes, we did not further investigate other advanced methods or plug-in a visual encoder, as our proposed strategies already significantly outperform the baseline.
>
> We appreciate the reviewer's consideration of alternative approaches and further enhancements. However, based on the experimental results and the performance achieved by our proposed methods, we are confident in the efficacy of our approach. We believe that the proposed method can serve as a strong and easy-to-apply baseline for the multimodal QA problem as well as a practical solution in real-world applications, and it remains open to future research endeavors that may explore additional possibilities.
>
> 2. Regarding the motivation of adopting “rationale”, we would like to clarify as follows.
>
> In specific, we aim to generate the rationale based on visual features and textual or tabular inputs to produce textual reasoning descriptions about the interrelations across different modalities. Since the ScienceQA dataset provides intermediate reasoning across image(visual) and question(texual) as “Explanation”, our rationale generator, fine-tuned on the ScienceQA benchmark, has the capability to perform cross-modal reasoning and describe the relationships between different modalities effectively.
>
> In figure 2, the image is the logo of Santa Anita Park, the text is about the origin of Santa Anita Park, and the table contains information about Race and Track. It is also challenging for humans to discover the correlation between three modalities, as the only common point among them is Santa Anita Park. In fact, this is related to the very low correlation between different modalities in MultimodalQA dataset. Besides, it provides only a limited number of cross-three-modal samples. Therefore, it would be difficult to present a cross-three-modal QA sample with high correlation between modalities. From a practical standpoint, the multimodal context provided by users may have high correlation. To this end, our purpose of introducing a rationale generator is to enhance reasoning across different modalities, particularly when there is a strong correlation between them. But due to the limitations of the existing datasets, we were unable to fully showcase this capability of our rationale generator in a satisfactory manner.
> However, there are more cases of cross-two-modal QA samples. We will add more examples for illustrating the cross-modal reasoning by the generated rationale.

---

### Official Review · Reviewer_htMw · 2023-08-07

**Soundness:** 3

**Excitement:**

3: Ambivalent: It has merits (e.g., it reports state-of-the-art results, the idea is nice), but there are key weaknesses (e.g., it describes incremental work), and it can significantly benefit from another round of revision. However, I won't object to accepting it if my co-reviewers champion it.

**Paper Topic And Main Contributions:**

The paper addresses the task of multimodal question answering (MMQA) by proposing a novel framework called UniMMQA. The paper focuses on integrating information from text, tables, and images into a text-to-text format to leverage the power of pre-trained language models effectively. While the novelty of the paper is acceptable, the proposed method appears to be relatively complex, which may raise concerns about its practicality and ease of implementation.

**Reasons To Accept:**

One strength of the paper is the attempt to tackle the challenging task of multimodal question answering by unifying text, tables, and images into a text-to-text format.  Unifying text, tables, and images into a text-to-text format is an innovative approach that allows the exploitation of pre-trained language models for this task. The use of diversified image captioning and position-enhanced table linearization techniques shows an effort to effectively leverage information from multiple modalities. The incorporation of a multimodal rationale generator to produce textual descriptions of cross-modal relations is a valuable addition, as it can improve the reasoning capabilities of the UniMMQA framework.

**Reasons To Reject:**

The main weakness of the paper is the potential complexity of the proposed UniMMQA framework. The introduction of multiple techniques, such as diversified image captioning, table linearization, and cross-modal rationale generation, might make the overall method intricate and difficult to implement for practitioners. The paper should include more comprehensive explanations and examples to clarify the integration process and make the framework more concise.

**Reproducibility:**

3: Could reproduce the results with some difficulty. The settings of parameters are underspecified or subjectively determined; the training/evaluation data are not widely available.

**Reviewer Confidence:**

4: Quite sure. I tried to check the important points carefully. It's unlikely, though conceivable, that I missed something that should affect my ratings.

---

> ### Author Rebuttal · Authors · 2023-08-28
>
> I would like to express my gratitude for your time and effort in reviewing our paper. We appreciate your valuable feedback and suggestions, which have prompted us to revisit our work and address the concerns raised in your evaluation. In this rebuttal, we aim to clarify certain points and provide further justification for the significance and contributions of our research.
>
> 1. Regarding the potential complexity of the proposed method, we would like to clarify as follows.
>
> it is essential to note that the introduction of multiple techniques, such as image captioning and table linearization, is fundamental to effectively address the challenges posed by the multi-modal Question Answering (MMQA) task. Given the multi-modal nature of input context in MMQA task, it is necessary to extract the features of each modality. Common methods for extracting image features include image encoder and image caption, while techniques such as position-enhanced encoding and template-based encoding are used for extracting table features. These techniques are widely adopted in the existing advanced models for MMQA. And we just made some improvements to them to make them more conducive to downstream QA tasks. Therefore, compared to many existing advanced methods, our framework does not introduce excessive complexity or intricacy.
>
> While the introduction of cross-modal rationale generation may add some potential complexity, we firmly believe that its benefits outweigh the potential drawbacks. Our introduction of this technique, which generates textual reasoning descriptions about the interrelations across different modalities, effectively helps downstream QA tasks. Most existing work often overlooks the correlation between modalities. By introducing cross-modal rationale generation, we provide an effective means to capture and utilize such correlations. To alleviate the complexity and difficulty of implementation for practitioners, we encapsulated these techniques, allowing users to easily incorporate the desired components by providing appropriate inputs. We also want to emphasize that we will be publishing our code soon, which will further validate and support our claims.
>
> 2. Regarding the comprehensive explanations and examples, we would like to clarify as follows.
>
> In our paper, Figure 2 and Case Study show comprehensive explanations and examples to clarify the integration process. Figure 2 illustrates the convertion of image and table inputs into text through the image captioning and table linearization respectively, and the rationale generator describe the relationships between different modalities. The output of these techniques are concatenated with question to serve as the input for the downstream QA model. Figure 4-6 in Case Study provide practical examples showcasing the usage of these three techniques, including the input and output of each techniques, as well as their impact on downstream QA task. By presenting these examples, we aim to provide readers with a clear understanding of how the techniques operate together and their implications for the overall framework.

---

### Official Review · Reviewer_T6GB · 2023-08-08

**Soundness:** 4

**Excitement:**

4: Strong: This paper deepens the understanding of some phenomenon or lowers the barriers to an existing research direction.

**Paper Topic And Main Contributions:**

The authors of this paper argue that current MMQA approaches are limited by relying solely on single-modal or bi-modal models. This limitation hinders their ability to effectively integrate information across all modalities and take advantage of pre-trained language models. To overcome these limitations, the authors propose a novel framework called UniMMQA. UniMMQA unifies three different input modalities into a text-to-text format using position-enhanced table linearization and diversified image captioning techniques. To further enhance cross-modal reasoning, a multimodal rationale generator is incorporated to produce textual descriptions of cross-modal relations for adaptation into the text-to-text generation process. Experimental results on three MMQA benchmark datasets demonstrate the superiority of UniMMQA in both supervised and unsupervised settings.

**Reasons To Accept:**

This paper is well-written and presents a straightforward approach to improving the performance of MMQA models by aggregating multiple modals. The idea of generating intermediate reasoning descriptions to seek connections and relationships between different modalities is particularly interesting. The results are solid and suggest that this approach has potential for improving performance on MMQA datasets. It remains to be seen how baselines such as MGT and SKURG will perform on ManymodalQA and MMConvQA.

**Reasons To Reject:**

NA

**Reproducibility:**

4: Could mostly reproduce the results, but there may be some variation because of sample variance or minor variations in their interpretation of the protocol or method.

**Reviewer Confidence:**

4: Quite sure. I tried to check the important points carefully. It's unlikely, though conceivable, that I missed something that should affect my ratings.

---

> ### Author Rebuttal · Authors · 2023-08-28
>
> Thank you for your valuable feedback and for acknowledging the quality of my paper. I sincerely appreciate the time and effort you have dedicated to reviewing my work.
>
> Regarding the performance of baselines such as MGT and SKURG on ManymodalQA and MMConvQA, we would like to clarify as follows.
>
> As for MGT, the code has not been made available as open-source at the time of our study. Therefore, we could only rely on the performance metrics reported in their paper. Regrettably, they didn’t conduct experiments on ManymodalQA and MMConvQA datasets. As for SKURG, they first built a knowledge graph to model the relations between multi-modal sources. Specifically, they extract entities from different sources and extract their relation based on certain attributes specific to MultimodalQA. Since these attributes are not provided in the ManymodalQA and MMConvQA datasets, it is challenging to adapt this knowledge graph construction method directly to these datasets.

---

### Meta-Review · Area_Chair_gL9x · 2023-09-19

**Recommendation:** 3

**Metareview:**

This work introduces a new framework, UniMMQA for multimodal QA (text, table, image), which unifies multimodal input into a text-to-text format by using table linearization and diversified image captioning techniques. On multimodalQA datasets, the proposed method outperforms prior approaches. While the method is solid and the experimental results are strong, I agree with the reviewer smpP on limited novelty---according to the author's response, the main technical novelties are combining TopP and TopK sampling at the decoding time and exploring different ways of linearizing table inputs, which sounds incremental to me as well.

---

### Decision · Program_Chairs · 2023-10-07

**Decision:**

Accept-Findings

**Comment:**

This work introduces a new framework, UniMMQA for multimodal QA (text, table, image), which unifies multimodal input into a text-to-text format by using table linearization and diversified image captioning techniques. On multimodalQA datasets, the proposed method outperforms prior approaches. While the method is solid and the experimental results are strong, I agree with the reviewer smpP on limited novelty---according to the author's response, the main technical novelties are combining TopP and TopK sampling at the decoding time and exploring different ways of linearizing table inputs, which sounds incremental to me as well.